# Genetic Contribution of Endometriosis to the Risk of Developing Hormone-Related Cancers

**DOI:** 10.3390/ijms22116083

**Published:** 2021-06-04

**Authors:** Aintzane Rueda-Martínez, Aiara Garitazelaia, Ariadna Cilleros-Portet, Sergi Marí, Rebeca Arauzo, Jokin de Miguel, Bárbara P. González-García, Nora Fernandez-Jimenez, Jose Ramon Bilbao, Iraia García-Santisteban

**Affiliations:** 1Department of Genetics, Physical Anthropology and Animal Physiology, Faculty of Medicine and Nursing, University of the Basque Country (UPV/EHU) and Biocruces-Bizkaia Health Research Institute, 48940 Leioa, Spain; arueda015@ikasle.ehu.eus (A.R.-M.); agaritazelaia001@ikasle.ehu.eus (A.G.); acilleros001@ikasle.ehu.eus (A.C.-P.); sergi.mari@ehu.eus (S.M.); rarauzo001@ikasle.ehu.eus (R.A.); jdemiguel008@ikasle.ehu.eus (J.d.M.); bgonzalez087@ikasle.ehu.eus (B.P.G.-G.); nora.fernandez@ehu.eus (N.F.-J.); joseramon.bilbao@ehu.eus (J.R.B.); 2Spanish Biomedical Research Center in Diabetes and Associated Metabolic Disorders (CIBERDEM), 28029 Madrid, Spain

**Keywords:** endometriosis, hormone-related cancers, epithelial ovarian cancer, breast cancer, endometrial cancer, mendelian randomization

## Abstract

Endometriosis is a common gynecological disorder that has been associated with endometrial, breast and epithelial ovarian cancers in epidemiological studies. Since complex diseases are a result of multiple environmental and genetic factors, we hypothesized that the biological mechanism underlying their comorbidity might be explained, at least in part, by shared genetics. To assess their potential genetic relationship, we performed a two-sample mendelian randomization (2SMR) analysis on results from public genome-wide association studies (GWAS). This analysis confirmed previously reported genetic pleiotropy between endometriosis and endometrial cancer. We present robust evidence supporting a causal genetic association between endometriosis and ovarian cancer, particularly with the clear cell and endometrioid subtypes. Our study also identified genetic variants that could explain those associations, opening the door to further functional experiments. Overall, this work demonstrates the value of genomic analyses to support epidemiological data, and to identify targets of relevance in multiple disorders.

## 1. Introduction

Endometriosis is a gynecological disorder affecting 190 million women worldwide. The disease is characterized by the presence of endometrial-like tissue in extra-uterine locations, causing pain and infertility [1]. Despite being a benign condition, growing epidemiological evidence shows that women with endometriosis could be at increased risk of developing certain types of hormone-related cancers, such as endometrial, breast or epithelial ovarian cancer [2,3,4,5].

Regarding the link between endometriosis and endometrial cancer, several reports describe that patients with endometriosis may be at higher risk for developing endometrial cancer during their lifetime [6,7]. Moreover, a recent study reported that women with endometriosis are more likely to be diagnosed with endometrial cancer at a younger age [8].

Data on the association between endometriosis and breast cancer are less consistent due to different reasons that have been reviewed elsewhere [9,10]. On the one hand, some studies provide evidence that women with endometriosis are more vulnerable to develop breast cancer, although they show contradicting results with regard to the age when this increased susceptibility is observed [11,12]. On the other hand, another group of studies are supportive of a negative association between endometriosis and breast cancer, the former possibly conferring protection from the latter [13,14].

Ovarian cancer comprises a histologically and genetically broad range of tumors of epithelial, sex cord-stromal and germ cell origin [15]. Epithelial ovarian cancer, which represents 90% of all ovarian tumors, has a robust relationship with endometriosis [16,17,18]. Among the multiple epithelial ovarian cancer histotypes, endometriosis has been more frequently linked to clear cell and endometrioid adenocarcinomas; in addition, literature regarding the link between endometriosis and the serous subtype suggests that there is no association between endometriosis and high-grade serous ovarian cancer, but there may be an association with low-grade serous ovarian cancer [19,20,21,22,23]. However, more studies are needed to draw definitive conclusions on whether these associations are indeed causal, and what is the biology behind them.

Even though a link between endometriosis and the three aforementioned hormone-related cancers exists, the underlying molecular mechanisms that explain this relationship are less understood. In this sense, some reports describe that endogenous estrogen levels [24] or immunological factors [25] might be involved in the malignant transformation of endometriosis into hormone-related carcinomas. Besides these local environmental factors, other studies suggest that genetic factors also play a pivotal role in the comorbidity between endometriosis and hormone-related tumors. A recent cross-disease genetic correlation and GWAS meta-analysis has described a moderate genetic correlation between endometriosis and endometrial cancer, providing evidence of pleiotropy for some associated genetic variants [26]. Similar reports have discovered *loci* shared between endometriosis and most ovarian cancer histotypes, suggesting that the epidemiological association between endometriosis and ovarian cancer is, at least in part, attributable to common genetic factors [27,28]. However, none of these genetic studies has explored whether, in addition to shared genetic markers, endometriosis and hormone-related cancers might also be causally linked through genetic factors using publicly available genetic data.

In this study, making use of public data from different genome wide association studies (GWAS), we evaluated the genetic relationship between endometriosis and each of those hormone-related cancers. Using genetic data, we assessed the potentially causal associations between endometriosis and the hormone-related cancers using a two-sample mendelian randomization (2SMR) strategy [29]. 2SMR is a statistical tool that helps to identify causal associations between an exposure trait (i.e., endometriosis) and an outcome trait (i.e., hormone-related cancers) of two independent populations by using single nucleotide polymorphisms (SNPs) to mimic a randomized control trial. Moreover, additional downstream tests extend our analysis to determine whether the exposure and the outcome are linked through horizontal pleiotropy or present a high degree of heterogeneity. Our results shed light on the shared genetic susceptibility between endometriosis and some of the analyzed hormone-related cancers, and provide additional evidence for a causal genetic influence of endometriosis on the development of epithelial ovarian cancer.

## 2. Results

### 2.1. 2SMR Analyses between Endometriosis and Hormone-Related Cancers Suggest a Potential Causal Relationship between Endometriosis and Ovarian Cancer

Endometriosis has been associated with the susceptibility to suffer from certain types of hormone-related cancers, including endometrial, breast and ovarian cancers. In order to determine whether there is causality underlying this association, we performed individual 2SMR analyses between endometriosis (exposure) and endometrial, breast or ovarian cancers (outcomes) using SNPs as genetic instruments. Table 1 summarizes the results from the 2SMR analysis for each cancer type. Each analysis was carried out using three different methods: inverse variance weighted (IVW), weighted median (WM) and Mendelian randomization Egger (MRE) (see Materials and Methods section for further details). None of the methods was able to find any significant causal relationship between endometriosis and breast cancer, and only the MRE method gave a potential association between endometriosis and endometrial cancer, suggesting that some of the selected instruments are pleiotropic. Interestingly, we found evidence of a significant causal relationship between endometriosis and ovarian cancer, not only with the IVW method (Beta = 0.251, SE = 0.051, *p*-value = 9.34 × 10^−7^), but also with the WM (Beta = 0.258, SE = 0.068, *p*-value = 1.37 × 10^−4^) and the MRE (Beta = 0.840, SE = 0.311, *p*-value = 3.09 × 10^−2^) methods. The consistency of the effect reported by multiple methods supports a causal relationship between endometriosis and ovarian cancer.

To validate our results, we replicated the analysis following the same criteria using an independent endometriosis GWAS dataset from the UK Biobank (UKBB, *ukb-b-10903*, “self-reported: endometriosis”) as exposure. As shown in Appendix A, even with the small number of selected instruments (3 SNPs in total), this analysis still produced significant results between endometriosis and ovarian cancer according to the IVW and WM methods. The consistency across the two independent 2SMR analyses strongly supports the potentially causal link between endometriosis and ovarian cancer.

In addition to the global 2SMR estimates, we also analyzed the role of selected instruments in each hormone-related cancer at the single-SNP level (Figure 1). In the case of endometrial and breast cancers, the SNPs showed both positive and negative beta values, proposing that the risk to suffer from endometriosis and those cancers could be attributed to different alleles of the same polymorphic sites. In turn, in the case of ovarian cancer, the vast majority of SNPs showed consistent positive effects, indicating that those SNP alleles that increase the susceptibility to endometriosis also predispose to suffer ovarian cancer. Interestingly, in spite of the heterogeneous results, rs12037376 was the most significantly associated SNP across the three different analyses, with a positive beta value for endometrial and ovarian cancer, but a negative beta value for breast cancer.

### 2.2. Sensitivity Analyses between Endometriosis and Hormone-Related Cancers Point to Heterogeneity with Breast Cancer and Horizontal Pleiotropy with Endometrial Cancer

Next, we further explored the genetic relationship between endometriosis and the three hormone-related cancers using different sensitivity analyses, including heterogeneity and horizontal pleiotropy tests (Table 2). We found evidence for substantial heterogeneity between endometriosis and breast cancer with both IVW and MRE (Cochran’s Q statistics for IVW = 28.34, degrees of freedom [df] = 8, *p*-value = 0.0004, and Cochran’s Q’ statistics for MRE = 28.12, df = 7, *p*-value = 0.0002). Similarly, the heterogeneity test between endometriosis and endometrial cancer gave significant results with the IVW method (Cochran’s Q statistics for IVW = 32.67, df = 8, *p*-value = 0.00007). A positive heterogeneity test is usually an indicator of potential violation of MR assumptions [29], and in this case show that the Wald estimates in the breast and endometrial cancer analyses are heterogeneous.

Importantly, in the case of endometrial cancer, the Egger intercept was −0.171 ± 0.042, and deviated significantly from zero (*p*-value = 0.0047). These results suggest that the genetic relationship between endometriosis and endometrial cancer, rather than causal, is more likely the result of significant horizontal pleiotropy.

Of note, there was no evidence of heterogeneity or horizontal pleiotropy between endometriosis and ovarian cancer, supporting the idea that the diseases might be linked through a causal relationship, and not biased by horizontal pleiotropy.

### 2.3. 2SMR Estimates between Endometriosis and Ovarian Cancer Subtypes Suggest a Stronger Causal Genetic Link between Endometriosis and Clear Cell and Endometrioid Histotypes

Our data show that the overall association between genetically predicted endometriosis and ovarian cancer is strongly significant (IVW *p*-value = 9.34 × 10^−7^, see Table 1). Given that endometriosis has been reported to be more strongly associated with certain subtypes of ovarian cancer, especially clear cell and endometrioid, we decided to perform a global 2SMR analysis using the different ovarian cancer histotype datasets as outcomes. Appendix A shows that, besides overall ovarian cancer, the top significant associations were obtained for the clear cell (IVW *p*-value = 3.38 × 10^−8^) and endometrioid (IVW *p*-value = 5.08 × 10^−4^) histotypes. The scatter plots in Figure 2 and Appendix A show the estimated effect sizes of the SNPs on both the exposure (endometriosis) and the outcomes (ovarian cancer subtypes). The ascending slopes in all plots are indicative of a positive correlation between endometriosis and those ovarian cancer subtypes. Overall, our data strongly suggest that the comorbidity between endometriosis and ovarian cancer, especially of clear cell and endometrioid subtypes, is mediated, at least in part, by a causal genetic relationship.

## 3. Discussion

Several epidemiological studies have reported comorbidity between endometriosis and certain hormone-related cancers [2,3,4,5], but the molecular mechanisms underlying those associations remain largely unexplored. Here, using public genetic data, we evaluate the potential causal or pleiotropic relationships between these pathologies employing a 2SMR approach.

Our 2SMR analysis explores the genetic link between endometriosis and hormone-related cancers and evaluates whether the associations observed could be causal or pleiotropic. Combining all the significant endometriosis risk SNPs as selected instruments (see Table 4), our overall 2SMR analyses reveal a distinct relationship in each case. Regarding endometriosis and endometrial cancer, our MR and sensitivity analyses provide evidence of heterogeneity and directional pleiotropy. These data indicate that the selected SNPs affect the outcome (endometrial cancer) through a pathway that is independent of the exposure (endometriosis). Our results are in line with a recently published genetic analysis where Painter et al. provided evidence for significant SNP pleiotropy between both diseases [26], and we discover some novel genetic variants, namely rs12037376 and rs10167914, that could play a role in both disorders. The concordance between the results from other groups and ours highlights the importance of performing additional sensitivity tests that not only evaluate the potential causality, but also the horizontal pleiotropy that is usually discarded as nuisance [30].

In the case of endometriosis and breast cancer, the Egger intercept was not significantly different from zero, suggesting a lack of horizontal pleiotropy. However, the heterogeneity tests yielded significant results using both the IVW and MRE methods, indicating substantial variation among the selected instruments. Of note, the global 2SMR estimates using all methods (IVW, WM and MRE) in the replication exposure dataset from the UKBB showed a negative beta value. These results, although not significant, are in accordance with the inverse association between endometriosis and breast cancer that has been observed in several epidemiological studies [13,14], and would further suggest that some genetic variants predisposing for the former would protect from the latter. However, given the few significant SNPs selected from the UKBB exposure dataset (see Table 4), these results must be interpreted with caution. Indeed, other reasons might explain the underlying biology of this possible protective effect; one of them is the long-term protection against breast cancer that young women on danazol/GnRH agonist treatment for endometriosis have [11].

The most compelling finding is that our 2SMR analysis provides strong evidence of a causal association between endometriosis and epithelial ovarian cancer. The consistency of the estimates across methods using the Sapkota et al. and the UKBB endometriosis GWAS as discovery and replication exposure datasets, respectively, strengthens this idea. Furthermore, the non-significant *p*-values in the heterogeneity and horizontal pleiotropy tests indicate that the MR assumptions are not violated. A more detailed MR analysis focused on all reported epithelial ovarian cancer histotypes as outcome supports the idea that endometriosis and clear cell and endometrioid ovarian cancers are more robustly associated by a mechanism that involves shared genetics. Of note, the low-grade serous ovarian cancer, whose association with endometriosis had been suggested in some studies, yielded non-significant results in our 2SMR analysis.

Single SNP MR analysis using endometriosis as exposure and ovarian cancer as outcome showed that the top significant SNPs are rs11674184 (2SMR *p*-value = 3.99 × 10^−3^) and rs12037376 (2SMR *p*-value = 2.05 × 10^−3^), both of them with positive beta values. The former is located within an intron of *GREB1* on chromosome 2. Publicly available gene expression data (https://gtexportal.org/home/; accessed in January 2021) indicate that this SNP is a splicing quantitative trait *loci* (sQTL) of *GREB1* in ovarian tissue with genome-wide significance (*p*-value = 3.6 × 10^−13^). Similarly, rs12037376, located on 1p36.12, is reported to be a sQTL of the long non-coding RNA *LINC000339* in breast-mammary tissue (*p*-value = 3.3 × 10^−7^) and an expression QTL (eQTL) of *WNT4* at a suggestive significance level (*p*-value = 0.05) in ovary. Genomic region 1p36.12 has been found to be associated with both endometriosis and ovarian carcinomas [28], and endometriosis risk alleles located within that *locus* have been described to act through inverse regulation of nearby genes such as *CDC42* and *LINC000339* [31]. To add more complexity, according to the Endometrial Tissue eQTL browser v2 (http://reproductivegenomics.com.au/shiny/eeqtl2/; accessed in January 2021), rs12037376, the top SNP in our single-SNP MR analysis between endometriosis and endometrial and breast cancers, is a strong cis-eQTL of *LINC000339* (*p*-value ≈ 10^−12^) in endometrial tissue from women with endometriosis [32]. Future studies should revisit our MR analyses to further explore the role that these individual genetic variants might play in the regulation of both endometriosis and hormone-related cancers. In particular, those future studies should address whether those SNPs that passed the significance threshold are actually the causative polymorphism, or rather mediate the association through a more complex regulatory mechanism involving, for instance, regulation of gene expression.

Finally, although our *in silico* analyses provide robust evidence on the genetic association between endometriosis and certain types of hormone-related cancers, especially epithelial ovarian cancer, the present study also has certain limitations. First, although the use of available public genomic data is an efficient way to generate new hypotheses with moderate costs, it also implies that the analyses must be adapted to the sample sizes and ethnic groups of the existing published studies. In our case, although the sample sizes of the available GWAS were large, our analysis used data from women of European ancestry and one must be cautious when generalizing these findings to other ethnic groups. Second, our results are still very preliminary, and should be interpreted with caution, pending further experimental work to validate our conclusions.

## 4. Materials and Methods

### 4.1. GWAS Data Sources

In this study, we utilized either selected instruments or full summary statistics from public GWAS on endometriosis and hormone-related cancers including endometrial, breast, and epithelial ovarian cancer (referred to as “ovarian cancer” in this article) available in *NHGRI-EBI*
*GWAS catalog* (https://www.ebi.ac.uk/gwas/; accessed in January 2021) and *IEU GWAS database* (https://gwas.mrcieu.ac.uk/; accessed in January 2021). Table 3 gives a more detailed description of each dataset.

For the 2SMR analysis, selected instruments for endometriosis to be used as exposure data were obtained either from the Sapkota et al. case-control study [33], or the UKBB cohort study [34] available from the *GWAS catalog* or *IEU GWAS database*, respectively. The study from Sapkota et al. is a meta-analysis of 11 independent endometriosis GWAS, the largest case-control GWAS meta-analysis on this disease performed to date, and was used as the discovery dataset; UKBB endometriosis data (*ukb-b-10903*, “self-reported: endometriosis”) were used to replicate the results obtained in the discovery analyses. Regarding the outcome data on hormone-related cancers, full summary statistics from either overall or different cancer subtypes (see Table 3) were obtained from the largest and most recent case-control GWAS published to date [35,36,37], available in the *IEU GWAS database*.

### 4.2. Two-Sample Mendelian Randomization (2SMR) Analysis

We conducted a two-sample mendelian randomization (2SMR) analysis using the *TwoSampleMR* R package, setting endometriosis as exposure and the different hormone-related cancers as outcomes. For the discovery analysis, we used the Sapkota et al. case-control GWAS as exposure dataset, and each of the hormone-related female cancer GWASs as independent outcome datasets. For the replication analysis, the UKBB cohort data was used as exposure, and the same hormone-related female cancers as outcomes. In the case of outcome data, we used each GWAS as an independent outcome. In addition to overall ovarian cancer, we also explored its main clinical subtypes [37,38].

The genetic instruments to be used as exposure data were queried in MR base [29] and formatted using the *format_data* or *extract_instruments* functions in *TwoSampleMR*. These functions extract SNPs that are strongly associated with the exposure. In the case of the GWAS by Sapkota et al., SNPs below the suggestive genome-wide significance value of 10^−5^ from the European Ancestry population were selected, and subsequently clumped into 11 independent instruments using the *clump_data* function (r^2^ < 0.001, window > 10,000 kb). In the case of the UKBB “Non-cancer illness code, self-reported: endometriosis” dataset, the 3 instruments selected were already clumped. Detailed information on the selected SNPs in exposure data is shown on Table 4.

We retrieved summary-level data for the association of the selected instruments and outcomes of interest using the *extract_outcome_data* function in *TwoSampleMR*, which generated an outcome data-frame. This data-frame contains, among others, the beta, standard error and *p*-values for the selected instruments that are present in the outcome GWAS (see Appendix A); if the particular SNP is no available, a proxy is used. The summary statistics for endometrial cancer were not available in the *IEU GWAS database* so they were downloaded from the *GWAS catalog* and formatted to the requirements of the *TwoSampleMR* package.

Both the exposure and outcome data-frames were harmonized using the *harmonise_data* function to remove ambiguous and/or palindromic SNPs, as it is the case for rs760794 and rs4762326, which were excluded from the analysis.

The final data-frame was analyzed with the *mr* function using different methods. To calculate the causal estimates between the exposure and the outcome, the inverse variance weighted (IVW) method was used. IVW is the simplest way to obtain a MR estimate using multiple SNPs, and treats each of them as a valid instrument. The fixed-effects meta-analysis framework employed in this paper assumes that each SNP provides the same estimate, or, in other words, that none of the SNPs exhibit horizontal pleiotropy. Additionally, we calculated causal estimates with two other methods: the weighted median (WM), and that the Mendelian randomization Egger (MRE) method. The WM is an alternative approach that takes the median effect of all available SNPs. This has the advantage that only 50% of the SNPs need to be valid instruments (no horizontal pleiotropy, no association with confounders, strong association with the exposure). It can be obtained by weighting the contribution of each SNP by the inverse variance association with the outcome. The MRE method relaxes the assumption of no horizontal pleiotropy, and adapts the IVW analysis by allowing a non-zero intercept and the net horizontal pleiotropic effect across all SNPs to be directional. Thus, it allows one or all of the SNPs used as instruments to be pleiotropic.

Scatter plots were generated using the *mr_scatter_plot* function. Additionally, the MR estimate of each individual SNP was obtained using *singlesnp_mr*, a function that calculates the Wald Ratio of every genetic variant, and was plotted using the *mr_forest_plot* function in *TwoSampleMR*.

We also performed additional sensitivity analyses. On the one hand, we carried out a heterogeneity test using the *mr_heterogeneity* function, and on the other hand, we calculated the MR-Egger intercept, which is an indication of the net directional pleiotropy, using the *mr_pleiotropy_test* function.

The R scripts generated for this analysis are available in the Appendix A.

## 5. Conclusions

Our study confirms the association between endometriosis and some hormone-related cancers and provides further evidence of a shared genetic etiology. Our 2SMR analysis further confirms that endometriosis and endometrial cancer are linked by a pleiotropic association, but gives inconclusive results about breast cancer, in line with previous literature. Importantly, we demonstrate that the association between endometriosis and ovarian cancer, especially of clear cell and endometrioid subtypes, is caused, at least in part, by a shared genetic origin. Up to date, it has been widely accepted that patients with endometriosis need to be regularly checked to prevent malignant transformation into certain hormone-related carcinomas, especially to epithelial ovarian cancer. Our results go beyond this idea, and propose that a common genetic makeup might also contribute, at least in part, to the increased malignancy.

## Figures and Tables

**Figure 1 ijms-22-06083-f001:**
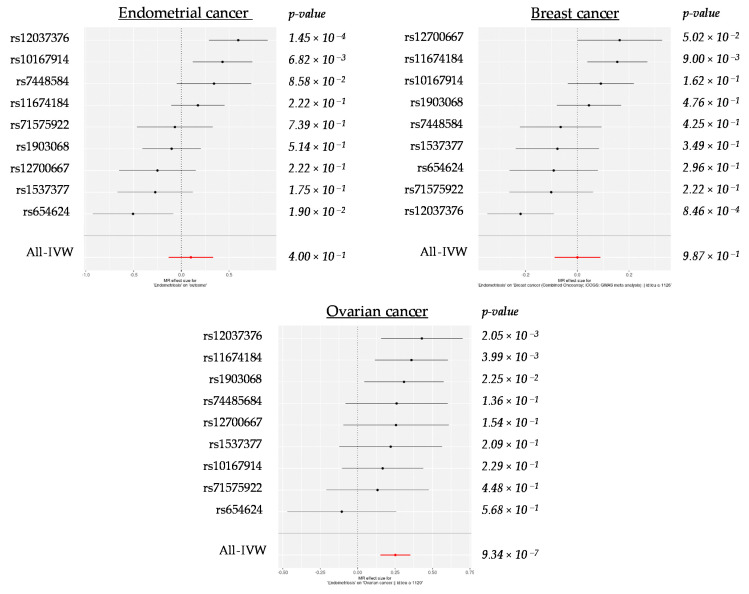
Forest plots showing beta (±standard error) and *p*-values of the single-SNP 2SMR analysis between endometriosis and endometrial, breast and ovarian cancers.

**Figure 2 ijms-22-06083-f002:**
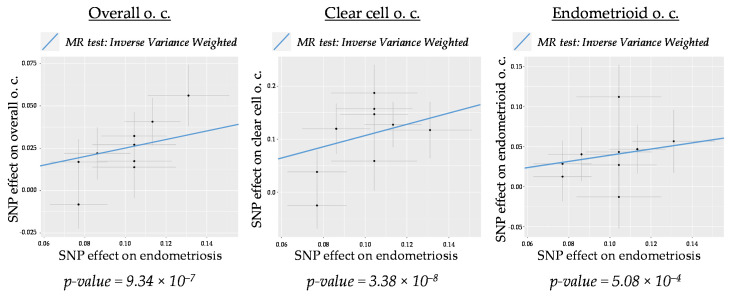
Scatter plots for 2SMR analyses of the causal effect of endometriosis on overall, clear cell and endometrioid ovarian cancer (o. c.). The slope of the line corresponds to the estimated MR effect (beta value) calculated with the inverse variance weighted method.

**Table 1 ijms-22-06083-t001:** Global 2SMR estimates between endometriosis endometrial, breast and ovarian cancers, using the Sapkota et al. endometriosis GWAS as exposure. Estimates were calculated using the inverse variance weighted (IVW), weighted median (WM), MR-Egger (MRE) methods. Beta (log Odds Ratio), standard error (SE) and *p*-values are indicated. Significant associations (*p*-value < 0.05) are highlighted in bold.

Outcome and Method	Beta	SE	*p*-Value
Endometrial cancer			
IVW	0.100	0.118	0.400
WM	0.028	0.093	0.767
**MRE**	**1.786**	**0.420**	**0.004**
Breast cancer			
IVW	0.001	0.045	0.987
WM	0.007	0.038	0.849
MRE	−0.068	0.294	0.824
Ovarian cancer			
**IVW**	**0.251**	**0.051**	**9.34 × 10^−7^**
**WM**	**0.258**	**0.068**	**1.37 × 10^−4^**
**MRE**	**0.840**	**0.311**	**3.09 × 10^−2^**

**Table 2 ijms-22-06083-t002:** Sensitivity tests between endometriosis and endometrial, breast and ovarian cancer, using the Sapkota et al. endometriosis GWAS as exposure. Significant associations (*p*-value < 0.05) are highlighted in bold. Q: Cochran’s Q statistic; df: degrees of freedom.

**Heterogeneity Test**			
**Outcome and Method**	**Q**	**Q_df**	***p*-Value**
Endometrial cancer			
**IVW**	**32.67**	**8**	**0.00007**
MRE	9.69	7	0.20671
Breast cancer			
**IVW**	**28.34**	**8**	**0.00041**
**MRE**	**28.12**	**7**	**0.00021**
Ovarian cancer			
IVW	7.12	8	0.52346
MRE	3.46	7	0.83951
**Horizontal Pleiotropy Test**			
**Outcome**	**Egger Intercept**	**SE**	***p*-Value**
Endometrial cancer	**−0.171**	**0.042**	**0.00472**

Breast cancer	0.007	0.029	0.81924

Ovarian cancer	−0.059	0.031	0.09719

**Table 3 ijms-22-06083-t003:** Description of GWAS used in each analysis.

Phenotype	Data Source	GWAS ID	Sample Size (Cases/Controls)	Population	1st Author, Year [Reference]
Endometriosis					
Discovery	GWAS catalog	GCST004549	208,641 (17,045/191,596)	European and Japanese ^1^	Sapkota, 2017 [33]
Replication	IEU GWAS db	ukb-b-10903	462,933 (3809/459,124)	European (UK)	UKBB cohort [34]
Endometrial cancer (e. c.)	GWAS catalog	GCST006464	121,885 (12,906/108,979)		O’Mara, 2018 [35]
Breast cancer (b. c.)	IEU GWAS db	ieu-a-1126	228,951 (122,977/105,974)	European	Michailidou, 2017 [36]
Ovarian cancer (o. c.) ^2^					Phelan, 2017 [37]
Overall o. c.	IEU GWAS db	ieu-a-1120	66,450 (25,509/40,941)	European	
High grade serous o. c.	IEU GWAS db	ieu-a-1121	53,978 (13,037/40,941)	European	
Low grade serous o. c.	IEU GWAS db	ieu-a-1122	41,953 (1012/40,941)	European	
Invasive mucinous o. c.	IEU GWAS db	ieu-a-1223	42,358 (1417/40,941)	European	
Clear cell o. c.	IEU GWAS db	ieu-a-1124	42,307 (1366/40,941)	European	
Endometrioid o. c.	IEU GWAS db	ieu-a-1125	43,751 (2810/40,941)	European	
High grade and low grade serous o. c.	IEU GWAS db	ieu-a-1228	54,990 (14,049/40,941)	European	
Serous o. c.: low grade and low malignant pot.	IEU GWAS db	ieu-a-1229	43,907 (2966/40,941)	European	
Serous o. c.: low malignant pot.	IEU GWAS db	ieu-a-1230	42,895 (1954/40,941)	European	
Mucinous o. c.: invasive and low malignant pot.	IEU GWAS db	ieu-a-1231	43,507 (2566/40,941)	European	
Low malignant potential mucinous o. c.	IEU GWAS db	ieu-a-1232	42,090 (1149/40,941)	European	
Low malignant potential o. c.	IEU GWAS db	ieu-a-1233	47,147 (3103/40,941)	European	

^1^ Only SNPs significant in European ancestry populations were considered for this study. ^2^ For simplicity, we refer to “epithelial ovarian cancer” as “ovarian cancer” through the manuscript.

**Table 4 ijms-22-06083-t004:** Summary information on endometriosis SNPs used as genetic instruments for the 2SMR analysis. SNP: single nucleotide polymorphism; Chr.: chromosome; TSS: transcription start site; EAF: effect allele frequency; SE: standard error. SNPs are ordered based on ascending *p*-values.

SNP	Effect/Other Allele	Chr.	Nearest TSS	EAF ^1^	Beta	SE	*p*-Value
Sapkota et al.							
rs11674184	G/T	2	GREB1	0.39	−0.113	0.014	3 × 10^−14^
rs12037376	A/G	1	WNT4	0.17	0.131	0.020	1 × 10^−12^
rs1903068	A/G	4	KDR	0.68	0.104	0.016	2 × 10^−11^
rs12700667	A/G	7	Intergenic	0.74	0.086	0.016	2 × 10^−8^
rs1537377	C/T	9	CDKN2B-AS1	0.40	0.077	0.014	2 × 10^−8^
rs71575922	G/C	6	SYNE1	0.16	0.104	0.021	2 × 10^−8^
rs74485684	T/C	11	FSHB	0.84	0.104	0.021	3 × 10^−8^
rs10167914	G/A	2	IL1A	0.30	0.104	0.018	5 × 10^−8^
rs760794	T/C	6	ID4	0.43	0.077	0.014	7 × 10^−8^
rs6546324	A/C	2	ETAA1	0.31	0.077	0.014	3 × 10^−7^
rs4762326	T/C	12	VEZT	0.47	0.068	0.014	1 × 10^−6^
UKBB							
rs61768001	C/T	1	*WNT4*	0.16	0.002	0.0003	1 × 10^−11^
rs9992737	T/C	4	*KDR*	0.28	−0.001	0.0002	2 × 10^−10^
rs11031005	C/T	11	*FSHB*	0.14	−0.002	0.0003	1.5 × 10^−9^

^1^ The effect allele frequency from the European population is displayed.

## Data Availability

No new data were created or analyzed in this study.

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
