# Peer review of "Genetic Contribution of Endometriosis to the Risk of Developing Hormone-Related Cancers"

_ijms, 2021, doi:10.3390/ijms22116083_

Round 1

Reviewer 1 Report

Useful study for prediction testing. Well prepared paper; clear with well designed statistical study.

Reviewer 2 Report

In this study authors investigated causal genetic association between endometriosis and hormone related gyn cancers. This manuscript provides interesting genetic background for causal relationship between ovarian cancer and endometriosis.  I just have one question; did authors look at causal relationship between endometriosis and type 1 endometrial cancer? Since type 1 is more hormonal related that would be interesting to see if the outcome would change by stratification. 

This manuscript is a resubmission of an earlier submission. The following is a list of the peer review reports and author responses from that submission.

Round 1

Reviewer 1 Report

The study applied an sound analysis by taking advantage of the already published vast data of endometriosis, endometrial cancers, ovary cancers and breast cancers, with the aim to assess if there is a common genetic susceptibility link between endometriosis and gynecological cancers. Taking advantage of these valuable WGAS data from various consortiums to explore new biology question is in generally a sound approach. As I am not particularly familiar with the GWAS analysis tools, methods, statistics, the appropriateness of these should be evaluated by lead experts to make sure that the conclusion is made based on solid and correct analysis methods and statistics. One particular comment that I would like to highlight is to include a section of limitations of the study. The finding is still very preliminary and will need substantial independent and experimental work to validate that the findings and support the conclusions. For those SNPs with significance, the SNPs might not necessary the causative mutation. Rather, it could be considered to see if there is a functional changes (e.g. gene expression, mutations) to the near TSS genes that lead to concordance development of endometriosis and gynecological cancers.

Other small comments:

The quality of Figure 3 should be improved.    

Reviewer 2 Report

Rueda-Martínez et al., investigate the genetic links between gynecological malignancies and endometriosis. The study is well justified and executed and provides insights that may form the basis for future functional investigations. I have only minor queries:

Major comments: 

  1. The authors identify SNPs from associated with ovarian cancer that are also linked to endometriosis. If the endometriosis GWAS is independently analysed do any of these SNPs reach genome-wide significance. If not, could the authors speculate as to why.

Minor comments:

  1. Thorough proofreading is required to correct grammatical and typographical errors.

Reviewer 3 Report

In the manuscript cancers-1110807, entitled “Genetic contribution of endometriosis to the risk of developing gynecological cancers” by Rueda-Martínez A et al., the authors assess the genetic contribution responsible for endometriosis and the most frequent gynecological tumors among women with endometriosis, such as breast, endometrial and ovarian cancer. The genetic analyses are well conducted and performed. The manuscript is clear and well-written. The presented data have the potential to understand the link between the individual genetic profile and the onset of specific diseases. Indeed, the reported genetic analyses represent a fundamental importance to medical and health care since they add a piece in how one pathological phenotype (in this case, endometriosis) is causally related to another (in this case, specific gynecological cancers).

However, some minor revisions are needed, and some comments are suggested:

  • The method of “Mendelian randomization” should be better explained in the main text (introduction, discussion) as well as the importance to use suitable tests for analyzing genetic heterogeneity and horizontal pleiotropy. See the paper by Hemani G et al. (Hum Mol Genet. 2018; doi: 10.1093/hmg/ddy163) and by Cho Y et al. (Nature Communications 2020; doi: 10.1038/s41467-020-14452-4).
  • The authors assert in the Discussion section that the most significantly associated regions in their study included the FGFR2, TOX3 and CCND1 genes, which had never been associated to endometriosis. Considering these genes, have the association studies been done or do they report negative results? Please, specify and discuss.
  • GWASs take the advantage to analyze susceptibility of genetic variants to a particular disease in wide populations, but it is important to underline the necessity to evaluate the specific risk allele in the pathogenesis of the disease in definite ethnic groups. Please, specify and discuss.
  • Some English sentences should be checked in the entire manuscript.
  • Row 87: “thosegynecological” must be fixed.
  • Table 1: FinnGen cohort (first row). Which year?

Reviewer 4 Report

The manuscript has two aims including: (1) to identify the SNPs that are associated with both endometriosis and the risk of  endometrioid, ovarian and breast cancers, and (2) to assess whether there is a causal relationship between endometriosis and the risk of those cancers using two-sample mendelian randomization (2SMR).

I appreciate the authors for doing a huge amount of analyses. I also understand that it is difficult to present several results in one manuscript. However, I believe that the manuscript could have been better written. Overarching issues include multiple hypothesis testing which is not addressed, lack of quality control information, and incorrect statements about the existing literature and thinking in this field. Further, breast cancer is not a gynecological cancer. Perhaps the authors want to frame this as “women’s cancers” or “hormone-related cancers”, but gynecological cancers is not correct. Overall, I think there is too much information presented in this paper – it is difficult to follow.

I have some concerns and comments to the authors.

Major concerns:

  1. The introduction is missing relevant literature (e.g., PMID: 22361336, PMID: 27885265, among others). This is particularly relevant because these manuscripts make the case for a causal relationship between endometriosis and ovarian cancer. While the use of MR is a different approach, suggesting, as this manuscript does, that there was no prior evidence for a causal relationship is not correct. Further, the authors state that this is the first manuscript to look at a genetic link between endometriosis and ovarian cancer. This is not correct. The authors have cited two articles that have done this (references #24 & #25). Broadly, the field is in agreement that endometriosis is the cell of original for clear cell and endometrioid ovarian cancers. In addition, the current preponderance of evidence suggests that there is no association between high-grade serous ovarian cancer and endometriosis, but there may be an association between low-grade serous ovarian cancer and endometriosis (PMID: 27325851, PMID: 22361336).
  2. For aim (1), I understand that the authors identified 4 GWAS from literature (with endometriosis and each of the cancers as the outcomes), and then selected 2,014 SNPs were significant to all the four outcomes. What is the value of just focusing on SNPs that were significantly associated to ALL endometriosis and the three cancers? I think it is worth to consider the common SNPs between endometriosis and each type of cancer.
  3. For aim (2), the authors concluded that endometriosis does not have a causal effect on endometrioid cancer. Instead, they are associated because they share common SNPs. I am concerned about this conclusion, given that the authors looked at only the SNPs that are shared between endometriosis and ALL three cancers (as in the above comment). It is possible that some SNPs were the common cause of endometriosis and epidemiological cancer only, but not the cause of ovarian or breast cancers. Some of these SNPs may have an effect on risk of epithelial ovarian cancer through its effect on endometriosis.
  4. The authors found a protective effect of endometriosis on breast cancer, although not significant. I am concerned that this effect is not a biological mechanism of endometriosis on breast cancer, given that many endometriosis patients may seek treatments. The effect of the treatments on cancer risk may cancel out the biological effect of endometriosis on cancer risk, which gives a spurious null or protective association. Therefore, I don’t think the conclusion in the manuscript is valid because the authors did not account for endometriosis treatments.
  5. The authors found a causal relationship between endometriosis and the risk of ovarian cancer, especially clear cell and endometrioid ovarian cancers. In the abstract page 1 lines 36-38, the authors claimed that “for the first time, we present robust evidence supporting a causal genetic association between endometriosis and ovarian cancer”. However, the authors did not show their estimate of this causal effect. They showed “betas” but they are confusing to me. Are they log of odds ratio? They authors presented different “betas” using different methods, but they did not make any judgement about which is the correct estimate, and how it is compared to the literature. My take-away message from the findings is that there is a causal effect of endometriosis on ovarian cancer, which is not a novel result.
    1. In the background (page 1, lines 67-68) the authors stated that the current evidence is not sufficient to draw a causal association between endometriosis and ovarian cancer. I would suggest the authors to present their literature review and to be specific about the reasons why they thought so. (See #1 above)
  6. The authors found 7,480 out of 11M SNPs that reached genome-wide significance. This is surprising and makes me wonder if quality control measures were taken into account in their use of the GWAS data. There are no details on this in the manuscript so it is difficult to assess.

Other comments

  1. I would suggest the authors to use epidemiological terms more correctly.
    1. I would not agree with the way the authors used the word “mediated”. For example, in the Abstract, page 1, lines 28-29, in the sentence “Since complex diseases are mediated by both environmental and genetic factors…”, I don’t think the word “mediated” is appropriate. I think “mediated” means that a factor is caused by the exposure and it in turn is a cause of the outcome. The same issue occurs in Page 1 line 39, and page 11 line 393.
    2. Based on Figure 2, I understand that the authors want to see the effect of endometriosis on each cancer, not the effect of SNPs on cancer. However, in the Results section page 7 lines 242-244, “…we also analyzed the role of selected instruments in each gynecological cancer at the single-SNP level (Figure 2). In the case of endometrial and breast cancers, the SNPs showed both positive and negative beta values…” So is it the effect of the SNPs on cancer risk, or the effect of endometriosis on cancer risk? Similar comments to lines 246-248.
    3. In Table 3, there is a column for “beta”. Is it the log of odds ratio of the association between endometriosis and the outcome? I would suggest the authors to be clearer about it.
    4. I don’t understand the claim about the effect directions in Results section page 6 lines 207-212: “We discovered that the effect directions were more consistent among ovarian cancer, breast cancer and endometriosis”. I would suggest the authors explain their statement with more details.
  2. Abstract, page 1 line 34: “the input of endometriosis was negligible when compared to the impact of the cancer studies”. This sentence is difficult to understand. I would suggest the authors to revise those sentences.
  3. Background section, page 1, lines 57-62. The authors mentioned that the association between endometriosis and risk of breast cancer is not consistent in the literature. I would suggest the authors to show their assessment why they are not consistent. Is it because of biases in the previous studies (and what bias), or is it because of the difference of the study populations?
  4. Background section, paragraphs 1 and 5, the authors said that in the literature the relationship between endometriosis and those cancers are clear and that endometriosis increases risk of those cancers. However, in paragraph 3, they mentioned that the association between endometriosis and risk of breast cancer is inconsistent in the literature. I would suggest the authors to make these sentences more consistent.
  5. The writing about number of SNPs that are significant and that are common between endometriosis and the risk of cancers is confusing (Methods page 4 second paragraph, and Results page 5 last paragraph and page 6 first paragraph). I would suggest the authors to make a figure showing how many SNPs are associated with each outcome, and how many of them are shared between endometriosis and each cancer.
  6. Methods section, Table 1, page 3: I would suggest the authors to present data based on the analysis (i.e. meta-analysis vs 2SMR) rather than by the type of disease.
  7. Table 1, page 3: The sample size is misleading. Although the numbers are very large, some of them come from case-control studies, Phelan 2017 for example, and majority of the participants were controls. In the analysis of low grade serous ovarian cancer for example, there were only about 1,000 cases and 40,000 controls. I would suggest the authors to show different sample sizes for cases and controls.
  8. Methods section page 3 GWAS data sources. There are several GWAS studies about the SNP-endometriosis and SNP-cancer risk associations in the literature. I would suggest the authors to explain why they chose those specific studies to include in the analyses.
  9. In the Materials and Methods page 4 lines 149-151: “Regarding the exposure data, the Sapkota et al. case-control GWAS was used as the discovery dataset and the UKBB cohort data for replication. In the case of outcome data, we used each GWAS as an independent outcome”. I assume that the authors mean: “To estimate the SNP-endometriosis association, we used the Sapkota et al. case-control study as the discovery dataset and the UKBB cohort data for replication. To estimate the SNP-cancer risk association, we used data from each GWAS described in table 1”. Am I correct? I would suggest the authors to rewrite those sentences.
  10. Results section, Table 3 shows different estimate effect of endometriosis on risk of cancer by method of calculation, i.e. inverse variance weighted, weighted mean, and MR-Egger (MRE). What does that mean? I understand each method has different assumptions. Which method do the authors think providing the most reliable results?
  11. The words in Figure 3 are so small to read. I would suggest the authors to make them bigger.

Reviewer 5 Report

General

This paper considered the potential association between endometriosis and endometrial, ovarian and breast cancer. It uses publically available GWAS data to: (1) try to identify SNPs associated with endometriosis AND the cancer types; and (2) determine whether certain SNPs associated with Endometriosis are also associated with the cancer types. The objective is to determine whether there is a common genetic basis for endometriosis and the other cancers.

1. Major

1. The first analysis appears to be driven by the meta analyses for the cancers. You conclude that you have identified new loci for endometriosis, but how certain are you about this? eg. What is effect size and 95%CI in the endometriosis only study. eg. Would you find the same results if endometriosis studies were not included at all. eg. If you included a set of SNPs from the other studies, and then did a false discovery analysis based on endometriosis data only, what would this be? Doesn't have to be these analyses, but seems some further justification of the conclusions seems needed for this part of the study is needed?

2. The 2SMR suggests a link to ovarian cancer in the "discovery" data. Why did you use only 3 of the instruments though in UK biobank? What about the others? What happens if you repeat the analysis using same instruments?

3. The paper describes the software used in detail, but not the methods therein. Please include the methods - software can change.

Minor

1. Can you provide a script for others to reproduce your analysis?

2. Table 1: how many cases / controls in each GWAS?

3. Lot of use of "significant", "significance" etc. Not always clear what you mean by this. Does this mean P<0.05? eg. line 155. Is this genome wide significance? Better to give p-value where needed, and avoid "significance" etc. Another reason: you do a lot of tests, comparisons etc, but do not adjust for multiple comparisons

4. Not sure what to make of the chi-square results line 207 onwards. What do they mean?

5. p6 line 226. Just because different methods that try to estimate the same thing give similar results does not necessarily strongly support a causal relationship. Suggest drop this language.

6. p8 line 272. Explain why you think no evidence of horizontal pleitropy. eg. Do you have sufficient precision to test?

7. Fig 3. SNP at the top right of first plot has weaker effects next two? Explain?

Reviewer 6 Report

Thank you. I have been asked to comment specifically on the MR analysis in this paper. I have not reviewed the details of the cross-trait GWAS.

In the introduction, the authors miss the point of two sample Mendelian randomisation when they talk about causal genetic associations. Mendelian randomisation uses genetic data to examine causal associations between an intermediate phenotype (here endometriosis) and disease outcomes (here cancers).

In the methods, some of the endometriosis SNPs are not GWAS significant. Furthermore, are these SNPs also associated with ovarian cancer? It would be useful to have that information in the table.

In the methods, the description of MR-Egger is not incorrect. MR-Egger allows the third assumption of instrument variable analysis to be relaxed and for one or all of the SNPs used in the instrument to be pleiotropic. The intercept from MR-Egger is an indication of the net directional pleiotropy.

In the results (section 3.2), they refer to SMR. The authors should stick to the term two-sample MR because SMR is a method that mainly relates to analyses using expression data and eQTLs. Rather than the two-sample approach that they have used here exploring causal relationships between complex traits.

In the results (section 3.2), they refer to genetic causality. There is not quite the same as what 2SMR is doing which is using genetics to infer causality between two traits.

In the results (section 3.3), the discussion of pleiotropy and heterogeneity is confused. Simply they have shown that the Wald estimates in the breast and endometrial cancer analyses are heterogeneous. For endometrial cancer, this heterogeneity balances out and there is not net direction pleiotropy. For breast cancer, there is and this means that the IVW estimate may be biased.

In the results (section 3.4), the term SMR should be avoided (see above). Again there is some confusion about what this MR approach is telling you about the causal relationship between endometriosis and ovarian cancer. It is not telling you about the genetic causal effect, it is using genetic data to infer an overall causal effect.